# ProtoAttend: Attention-Based Prototypical Learning

## Abstract

We propose a novel inherently interpretable machine learning method that bases decisions on few relevant examples that we call *prototypes*. Our method, ProtoAttend, can be integrated into a wide range of neural network architectures including pre-trained models. It utilizes an attention mechanism that relates the encoded representations to samples in order to determine prototypes. The resulting model outperforms state of the art in three high impact problems without sacrificing accuracy of the original model: (1) it enables high-quality interpretability that outputs samples most relevant to the decision-making (i.e. a sample-based interpretability method); (2) it achieves state of the art confidence estimation by quantifying the mismatch across prototype labels; and (3) it obtains state of the art in distribution mismatch detection. All this can be achieved with minimal additional test time and a practically viable training time computational cost.

## 1 Introduction

Deep neural networks have been pushing the frontiers of artificial intelligence (AI) by yielding excellent performance in numerous tasks, from understanding images (He et al., 2016) to text (Conneau et al., 2016). Yet, high performance is not always a sufficient factor - as some real-world deployment scenarios might necessitate that an ideal AI system is 'interpretable', such that it builds trust by explaining rationales behind decisions, allow detection of common failure cases and biases, and refrains from making decisions without sufficient confidence. In their conventional form, deep neural networks are considered as *black-box models* – they are controlled by complex nonlinear interactions between many parameters that are difficult to understand. There are numerous approaches, (Kim et al., 2018; Erhan et al., 2009; Zeiler & Fergus, 2013; Simonyan et al., 2013), that bring post-hoc explainability of decisions to already-trained models. Yet, these have the fundamental limitation that the models are not designed for interpretability. There are also approaches on the redesign of neural networks towards making them inherently-interpretable, as in this paper. Some notable ones include sequential attention (Bahdanau et al., 2015), capsule networks (Sabour et al., 2017), and interpretable convolutional filters (Zhang et al., 2018).

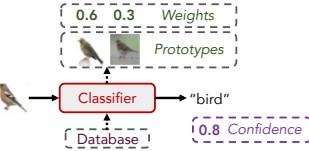

Figure 1: ProtoAttend bases the decision on a few prototypes from the database. This enables interpretability of the prediction (by visualizing the highest weight prototypes) and confidence estimation for the decision (by measuring agreement across prototype labels).

We focus on inherently-interpretable deep neural network modeling with the foundations of *prototypical learning*. Prototypical learning decomposes decision making into known samples (see Fig. 1), referred here as prototypes. We base our method on the principle that prototypes should constitute *a minimal subset of samples with high interpretable value that can serve as a distillation or condensed view of a dataset* (Bien & Tibshirani, 2012). Given that the number of objects a human can interpret is limited (Miller, 1956), outputting few prototypes can be an effective approach for humans to understand the AI model behavior. In addition to such interpretability, prototypical learning:

(1) provides an efficient confidence metric by measuring mismatches in prototype labels, allowing performance to be improved by refraining from making predictions in the absence of sufficient confidence, (2) helps detect deviations in the test distribution by measuring mismatches in prototype labels that represent the support of the training dataset, and (3) enables performance in the high label noise regime to be improved by controlling the number of selected prototypes. Given these motivations, prototypes should be controllable in number, and should be perceptually relevant to the input in explaining the decision making task. Prototype selection in its naive form is computationally expensive and perceptually challenging (Bien & Tibshirani, 2012). We design ProtoAttend to address this problem in an efficient way. Our contributions can be summarized as follows:

1. We present principles that can be guiding for the design of inherently-interpretable models based on sample-based interpretability.
2. We propose a novel method, ProtoAttend, for selecting input-dependent prototypes based on an attention mechanism between the input and prototype candidates. ProtoAttend is model-agnostic and can even be integrated with pre-trained models.
3. ProtoAttend allows interpreting the contribution of each prototype via the attention outputs.
4. For a 'condensed view', we demonstrate that sparsity in weights can be efficiently imposed via the choice of the attention normalization and additional regularization.
5. On image, text and tabular data, we demonstrate the four key benefits of ProtoAttend: interpretability, confidence control, diagnosis of distribution mismatch, and robustness against label noise. ProtoAttend yields superior quality for sample-based interpretability, better-calibrated confidence scoring, and more sensitive out-of-distribution detection compared to alternative approaches.
6. ProtoAttend enables all these benefits via the same architecture and method, while maintaining comparable overall accuracy.

## 2   RELATED WORK

**Prototypical learning:**  The principles of ProtoAttend are inspired by (Bien & Tibshirani, 2012). They formulate prototype selection as an integer program and solve it using a greedy approach with linear program relaxation. It seems unclear whether such approaches can be efficiently adopted to deep learning. (Chen et al., 2018) and (Li et al., 2018) introduce a prototype layer for interpretability by replacing the conventional inner product with a distance computation for perceptual similarity. In contrast, our method uses an attention mechanism to quantify perceptual similarity and can choose input-dependent prototypes from a large-scale candidate database. (Yeh et al., 2018) decomposes the prediction into a linear combination of activations of training points for interpretability using represent values. The linear decomposition idea also exists in ProtoAttend, but the weights are learned via an attention mechanism and sparsity is encouraged in the decomposition. In (Koh & Liang, 2017), the training points that are the most responsible for a given prediction are identified using influence functions via oracle access to gradients and Hessian-vector products.

**Metric learning:**  Metric learning aims to find an embedding representation of the data where similar data points are close and dissimilar data pointers are far from each other. ProtoAttend is motivated by efficient learning of such an embedding space which can be used to decompose decisions. Metric learning for deep neural networks is typically based on modifications to the objective function, such as using triplet loss and N-pair loss (Sohn, 2016; Cui et al., 2016; Hoffer & Ailon, 2014). These yield perceptually meaningful embedding spaces yet typically require a large subset of nearest neighbors to avoid degradation in performance (Cui et al., 2016). (Kim et al., 2018) proposes a deep metric learning framework which employs an attention-based ensemble with a divergence loss so that each learner can attend to different parts of the object. Our method has metric learning capabilities like relating similar data points, but also performs well on the ultimate supervised learning task.

**Attention-based few-shot learning:**  Some of our inspirations are based on recent advances in attention-based few-shot learning. In (Vinyals et al., 2016), an attention mechanism is used to relate an example with candidate examples from a support set using a weighted nearest-neighbor classifier applied within an embedding space. In (Ren et al., 2018), incremental few-shot learning is implemented using an attention attractor network on the encoded and support sets. In (Snell et al., 2017), a non-linear mapping is learned to determine the prototype of a class as the mean of its support set in the embedding space. During training, the support set is randomly sampled to mimic the inference task. Overall, the attention mechanism in our method follows related principles but fundamentally differs in that few-shot learning aims for generalization to unseen classes whereas the goal of our method is robust and interpretable learning for seen classes.

**Uncertainty and confidence estimation:** ProtoAttend takes a novel perspective on the perennial problem of quantifying how much deep neural networks' predictions can be trusted. Common approaches are based on using the scores from the prediction model, such as the probabilities from the softmax layer of a neural network, yet it has been shown that the raw confidence values are typically poorly calibrated (Guo et al., 2017). Ensemble of models (Lakshminarayanan et al., 2017) is one of the simplest and most efficient approaches, but significantly increases complexity and decreased interpretability. In (Papernot & McDaniel, 2018), the intermediate representations of the network are used to define a distance metric, and a confidence metric is proposed based on the conformity of the neighbors. (Jiang et al., 2018), proposes a confidence metric based on the agreement between the classifier and a modified nearest-neighbor classifier on the test sample. In (DeVries & Taylor, 2018), direct inference of confidence output is considered with a modified loss. Another direction of uncertainty and confidence estimation is Bayesian neural networks that return a distribution over the outputs (Kendall & Gal, 2017b) (Mullachery et al., 2018) (Kendall & Gal, 2017a).

## 3 PROTOATTEND: ATTENTION-BASED PROTOTYPICAL LEARNING

Consider a training set with samples, $\mathcal{T} = \{\mathbf{x_i}, y_i\}$. Conventional supervised learning aims to learn a model $s(\mathbf{x_i}; \mathbf{S})$ that minimizes a predefined loss $1/B \cdot \sum_{i=1}^{B} L(y_i, \hat{y}_i = s(\mathbf{x_i}; \mathbf{S}))^1$ at each iteration, where $B$ is the batch size for training. Our goal is to impose that decision making should be based on only a small number of training examples, i.e. *prototypes*, such that their linear superposition in an embedding space can yield the overall decision and the superposition weights correspond to their importance. Towards this goal, we propose defining a solution to prototypical learning with the following six principles:

i. $\mathbf{v_i} = f(\mathbf{x_i}; \theta)$ encodes all relevant information of $\mathbf{x_i}$ for the final decision. $f()$ considers the global distribution of the samples, i.e. learns from all $\{\mathbf{x_i}, y_i\}$. Although all the information in training dataset is embodied in the weights of the encoder[2], we construct the learning method in such a way that decision is dominated by the prototypes with high weights.

ii. From the encoded information, we can find a decision function so that the mapping $g(\mathbf{v_i}; \eta)$ is close to the ground truth $y_i$, in a consistent way with conventional supervised learning.

iii. Given candidates $\mathbf{x_j^{(c)}}$ to select the prototypes from, there exists weights $p_{i,j}$ (where $p_{i,j} \geq 0$ and $\sum_{j=1}^{D} p_{i,j} = 1$), such that the decision $g(\sum_{j=1}^{D} p_{i,j} \mathbf{v_j^{(c)}}; \eta)$ (where $\mathbf{v_j^{(c)}} = f(\mathbf{x_j^{(c)}}; \theta)$) is close to the ground truth $y_i$.

iv. When the linear combination $\sum_{j=1}^{D} p_{i,j} \mathbf{v_j^{(c)}}$ is considered, prototypes with higher weights $p_{i,j}$ have higher contribution in the decision $g(\sum_{j=1}^{D} p_{i,j} \mathbf{v_j^{(c)}}; \eta)$.

v. The weights should be sparse – only a controllable amount of weights $p_{i,j}$ should be non-zero. Ideally, there exists an efficient mechanism for outputting $p_{i,j}$ to control the sparsity without significantly affecting performance.

vi. The weights $p_{i,j}$ depend on the relation between input and the candidate samples, $p_{i,j} = r(\mathbf{x_i}, \mathbf{x_j^{(c)}}; \mathbf{\Gamma})$, based on their perceptual relation for decision making. We do not introduce any heuristic relatedness metric such as distances in the representation space, but we allow the model to learn the relation function that helps the overall performance.

Learning involves optimization of the parameters $\theta, \mathbf{\Gamma}, \eta$ of the corresponding functions. If the proposed principles (such as reasoning from the linear combination of embeddings or assigning relevance to the weights) are not imposed during training but only at inference, a high performance cannot be obtained due to the train-test mismatch, as the intermediate representations can be learned in an arbitrary way without any necessities to satisfy them.[3] The subsequent section presents ProtoAttend and training procedure to implement it.

### 3.1 NETWORK ARCHITECTURE AND TRAINING

The principles above are conditioned on efficient learning of an encoding function to encode the relevant information for decision making, a relation function to determine the prototype weights, and

---

[1]$\mathbf{S}$ represents the trainable parameters for $s(; \mathbf{S})$ and is sometimes not show for notation convenience.

[2]Training of $f()$ may also involve initializing with pre-trained models or transfer learning.

[3]For example, commonly-used distance metrics in the representation spaces fail at determining perceptual relevance between samples when the model is trained in a vanilla way (Sitawarin & Wagner, 2019).

a final decision making block to return the output. Conventional supervised learning comprises the encoding and decision blocks. On the other hand, it is challenging to design a learning method with a relation function with a reasonable complexity. To this end, we adapt the idea of attention (Corbetta & Shulman, 2002; Vaswani et al., 2017), where the model focuses on an adaptive small portion of input while making the decision. Different from conventional employment of attention in sequence or visual learning, we propose to use attention at sample level, such that the attention mechanism is used to determine the prototype weights by relating the input and the candidate samples via alignment of their keys and queries. Fig. 2 shows the proposed architecture for training and inference. The three main blocks are described below:

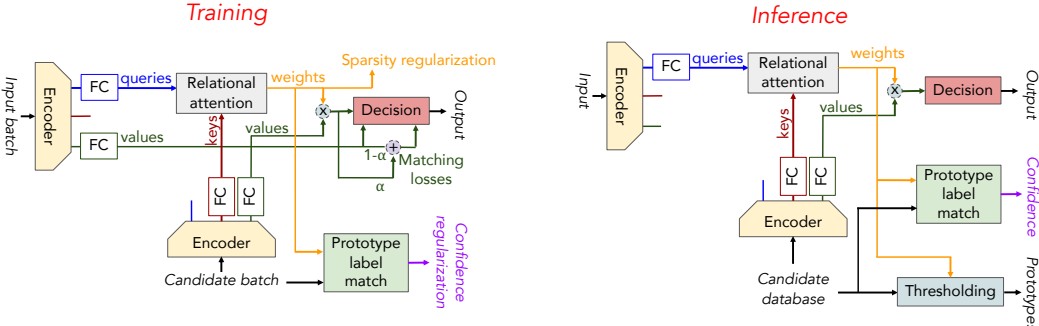

Figure 2: ProtoAttend method for training and testing. Shared encoder between input samples and the candidate samples generates input representations, that are mapped to key, query and value embeddings (with a single nonlinear layer). The alignment between keys and queries determines the weights of the prototypes, and the linear combination of the values determines the final decision. Conformity of the prototype labels is used as a confidence metric.

**Encoder:** A trainable encoder is employed to transform $B$ input samples (note that $B$ may be 1 at inference) and $D$ samples from the database of prototype candidates (note that $D$ may be as large as the entire training dataset at inference) into keys, queries and values. The encoder is shared and jointly updated for the input samples and prototype candidate database, to learn a common representation space for the values. The encoder architecture can be based on any trainable discriminative feature mapping function, e.g. ResNet (He et al., 2016) for images, with the modification of generating three types of embeddings. For mapping of the last encoder layer to key, query and value embeddings, we simply use a single fully-connected layer with a nonlinearity, separately for each.[4] For input samples, $\mathbf{V} \in \Re^{B \times d_{out}}$ and $\mathbf{Q} \in \Re^{B \times d_{att}}$ denote the values and queries, and for candidate database samples $\mathbf{K}^{(\mathbf{c})} \in \Re^{D \times d_{att}}$ and $\mathbf{V}^{(\mathbf{c})} \in \Re^{D \times d_{out}}$ denote the keys and values. For keys and queries, we use separate representations as the entire system is not symmetric, there are a lot of candidate samples and the model may prefer to learn the keys to arrange the representation space such that it is meaningful when their inner products with a single query are considered.

**Relational attention:** The relational attention yields the weight between the $i^{th}$ sample and $j^{th}$ candidate, $p_{i,j}$, via alignment of the corresponding key and query in dot-product attention form[5]:

$$p_{i,j} = n \left( \mathbf{K}^{(\mathbf{c})}_{\mathbf{j}} \mathbf{Q}_{\mathbf{i}}^{T} / \sqrt{d_{att}} \right), \tag{1}$$

where $n()$ is a normalization function to satisfy $p_{i,j} \geq 0$ and $\sum_{j=1}^{D} p_{i,j} = 1$ for which we consider softmax and sparsemax (Martins & Astudillo, 2016)[6]. The choice of the normalization function is an efficient mechanism to control the sparsity of the prototype weights, as demonstrated in experiments. Note that the relational attention mechanism does not introduce any extra trainable parameters.

**Decision making:** The final decision block simply consists of a linear mapping from a convex combination of values that results in the output $y_i$. Consider the convex combination of value embeddings, parameterized by $\alpha$:

$$\hat{y}_i(\alpha) = g \left( (1-\alpha)\mathbf{v_i} + \alpha \sum_{j=1}^{D} p_{i,j} \mathbf{v}_{\mathbf{j}}^{(\mathbf{c})} \right). \tag{2}$$

---

[4]There are other viable options for the mapping but we restrict it to a single layer to minimize the additional number of trainable parameters, which becomes negligible in most cases.

[5]We use $\mathbf{A_i}$ to denote the $i^{th}$ row of $\mathbf{A}$.

[6]Sparsemax encourages sparsity by mapping the Euclidean projection onto the probabilistic simplex.

For $\alpha = 0$, $L(y_i, \hat{y}_i(0))$ is the conventional supervised learning loss (ignoring the relational attention mechanism) that can only impose principles (i) and (ii), but not the principles (iii)-(vi). A high accuracy for $\hat{y}_i(0)$ merely indicates that the value embedding space represents each input sample accurately. For $\alpha = 1$, $L(y_i, \hat{y}_i(1))$ encourages the principles (i), (iii)-(iv), but not the principles (ii) and (vi).[7] A high accuracy for $\hat{y}_i(1)$ indicates that the linear combination of value embeddings accurately maps to the decision. For (vi), we propose that there should be a similar output mapping for the input and prototypes, for which we encourage high accuracy for both $\hat{y}_i(0)$ and $\hat{y}_i(1)$ with a loss term that is a mixture of $L(y_i, \hat{y}_i(0))$ and $L(y_i, \hat{y}_i(1))$ or guidance with an intermediate term, as $\hat{y}_i(0.5)$, is required. Lastly, when $\alpha \leq 0.5$, we obtain the condition that the input sample itself has the largest contribution in the linear combination. Intuitively, the sample itself should be more relevant for the output compared to other samples, so the principles (iii) and (iv) can be encouraged. We propose and compare different training objective functions in Table 1. We observe that the last four are all viable options as the training objective, with similar performance. We choose the last one for the rest of the experiments, as in some cases, slightly better prototypes are observed qualitatively (see Sect. 5.2 for further discussion).

Table 1: Ablation study. Impact of various training losses on ProtoAttend with softmax attention for Fashion-MNIST. $1 \leq i \leq N_t$ is the training iteration index and $N_t$ is the total number of iterations.

| Training objective function | Acc. % for $\hat{y}_i(0)$ | Acc. % for $\hat{y}_i(1)$ | $-\underset{\hat{y}=y}{E}\{C\}$ | $-\underset{\hat{y}\neq y}{E}\{C\}$ |
|---|---|---|---|---|
| $L(y_i, \hat{y}_i(0))$ | 94.28 | 13.13 | 0.029 | 0.194 |
| $L(y_i, \hat{y}_i(1))$ | 10.92 | 94.21 | 0.103 | 0.002 |
| $L(y_i, \hat{y}_i(0.5))$ | 94.01 | 94.25 | 0.927 | 0.049 |
| $L(y_i, \hat{y}_i(0)) + L(y_i, \hat{y}_i(1))$ | 94.37 | 94.38 | 0.931 | 0.047 |
| $(1 - i/N_t) \cdot L(y_i, \hat{y}_i(0)) +$ $(i/N_t) \cdot L(y_i, \hat{y}_i(1))$ | 94.14 | 94.18 | 0.927 | 0.049 |
| $L(y_i, \hat{y}_i(0)) + L(y_i, \hat{y}_i(1)) +$ $L(y_i, \hat{y}_i(0.5))$ | 94.37 | 94.45 | 0.928 | 0.047 |

To control the sparsity of the weights (beyond the choice of the attention operation), we also propose a sparsity regularization term with a coefficient $\lambda_{sparse}$ in the form of entropy, $L_{sparse}(\mathbf{p}) = -1/B \sum_{i=1}^{B} \sum_{j=1}^{D} p_{i,j} \log(p_{i,j} + \epsilon)$, where $\epsilon$ is a small number for numerical stability. $L_{sparse}(\mathbf{p})$ is minimized when $\mathbf{p}$ has only 1 non-zero value.

## 3.2 CONFIDENCE SCORING USING PROTOTYPES

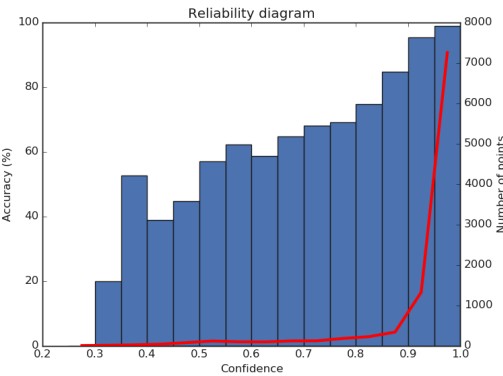

Figure 3: Impact of confidence on ProtoAttend accuracy. Reliability diagram for Fashion-MNIST, as in (Papernot & McDaniel, 2018). Bars (left axis) indicate the mean accuracy of predictions binned by confidence; the red line (right axis) shows the number of samples across bins.

ProtoAttend provides a linear decomposition (via value embeddings) of the decision into prototypes that have known labels. Ideally, labels of the prototypes should all be the same as the labels of the

---

[7]For example, simply assigning non-zero weights to another predetermined class, prototypical learning method can obtain perfect accuracy, but the assignment of predetermined class would be arbitrary.

input. When prototypes with high weights belong to the same class, the model shall be more confident and a correct classification result is expected, whereas in the cases of disagreement between prototype labels, the model shall be less confident and the likelihood of a wrong prediction is higher. With the motivation of separating correct vs. incorrect decisions via its value, we propose a confidence score based on the agreement between the prototypes:

$$C_i = \sum_{j=1}^{D} p_{i,j} \cdot I(y_j^{(c)} = \hat{y}_i), \tag{3}$$

where $I()$ is the indicator function. Table 1 shows the significant difference of the average confidence metric between correct vs. incorrect classification cases for the test dataset, as desired. In Fig. 3, the impact of confidence on accuracy is further analyzed with the reliability diagram as in (Papernot & McDaniel, 2018). When test samples are binned according to their confidence, it is observed that the bins with higher confidence yield much higher accuracy. There are small number of samples in the bins with lower confidence, and those tend to be the incorrect classification cases. In Section 4.4, the efficacy of confidence score in separating correct vs. incorrect classification is experimented in confidence-controlled prediction setting, demonstrating how much the prediction accuracy can be improved by refraining from small number of samples with low confidence at test time.

To further encourage confidence during training, we also consider a regularization term $L_{conf}(\mathbf{p}) = -1/B \sum_{i=1}^{B} \sum_{j=1}^{D} p_{i,j} \cdot I(y_j^{(c)} = y_i)$ with a coefficient $\lambda_{conf}$. $L_{conf}$ is minimized when all prototypes with $p_{i,j} > 0$ are from the same ground truth class with output $y_i$.[8]

## 4 EXPERIMENTS

### 4.1 SETUP

We demonstrate the results of ProtoAttend for image, text and tabular data classification problems with different encoder architectures (see Supplementary Material for details). Outputs of the encoders are mapped to queries, keys and values using a fully-connected layer followed by ReLU. For values, layer normalization (Lei Ba et al., 2016) is employed for more stable training. A fully-connected layer is used in the decision making block, yielding logits for determining the estimated class. Softmax cross entropy loss is used as $L()$. Adam optimization algorithm is employed (Kingma & Ba, 2014) with exponential learning rate decay (with parameters optimized on a validation set). For image encoding, unless specified, we use the standard ResNet model (He et al., 2016). For text encoding, we use the very deep convolutional neural network (VDCNN) (Conneau et al., 2016) model, inputting sequence of raw characters. For tabular data encoding, we use an LSTM model (Hochreiter & Schmidhuber, 1997), which inputs the feature embeddings at every timestep. See Supplementary Material for implementation details, additional results and discussions.

### 4.2 SPARSE EXPLANATIONS OF DECISIONS

We foremost demonstrate that our inherently-interpretable model design does not cause significant degradation in performance. Table 2 shows the accuracy and the median number of prototypes required to add up to a particular portion of the decision[9] for different prototypical learning cases. In all cases, very small accuracy gap is observed with the baseline encoder that is trained in conventional supervised learning way. The attention normalization function and sparsity regularization are efficient mechanisms to control the sparsity – the number of prototypes required is much lower with sparsemax attention compared to softmax attention and can be further reduced with sparsity regularization (see Supplementary Material for details). With a small decrease in performance, the number of prototypes can be reduced to just a handful.[10] There is difference between datasets, as intuitively expected from the discrepancy in the degree of similarity between the intra-class samples.

---

[8]Note that the gradients of this regularization term with respect to $p_{i,j}$ is either 0 or 1 and it is often insufficient to train the model itself from scratch. But it is observed to provide further improvements in some cases.

[9]E.g. if the prototype weights are [0.1, 0.15, 0.05, 0.25, 0.1, 0.05, 0.28, 0.02], then 2 prototypes are required for 50% of the decision, 6 for 90% and 7 for 95%.

[10]We observe that excessively high sparsity (to yield 1-2 prototypes in most cases) may sometimes decrease the quality of prototypes due to overfitting to discriminative features that are less perceptually meaningful.

Table 2: ProtoAttend achieves interpretability without significant degradation in performance. Accuracy and median number of prototypes to add up to 50%, 90% and 95% of the decision, quantified with prototype weights.

| Dataset | Method | Acc. % | No. of prototypes | | |
|---|---|---|---|---|---|
| | | | 50 % | 90 % | 95 % |
| MNIST | Baseline enc. | 99.70 | | - | |
| | Softmax attn. | 99.66 | 365 | 1324 | 1648 |
| | Sparsemax attn. | 99.69 | 2 | 4 | 5 |
| Fashion-MNIST | Baseline enc. | 94.74 | | - | |
| | Softmax attn. | 94.42 | 712 | 2320 | 2702 |
| | Sparsemax attn. | 94.42 | 4 | 10 | 11 |
| | Sparsemax attn. + sparsity reg. | 94.47 | 1 | 2 | 2 |
| CIFAR-10 | Baseline enc. | 91.97 | | - | |
| | Softmax attn. | 91.69 | 317 | 1453 | 1898 |
| | Sparsemax attn. | 91.44 | 5 | 14 | 16 |
| | Sparsemax attn. + sparsity reg. | 91.26 | 2 | 3 | 4 |
| DBPedia | Baseline enc. | 98.25 | | - | |
| | Softmax attn. | 98.20 | 63 | 190 | 225 |
| | Sparsemax attn. | 97.74 | 2 | 4 | 4 |
| Income | Baseline enc. | 85.68 | | - | |
| | Softmax attn. | 85.64 | 2263 | 9610 | 12419 |
| | Sparsemax attn. | 85.58 | 20 | 57 | 67 |
| | Sparsemax attn. + sparsity reg. | 85.41 | 3 | 6 | 7 |

(a) MNIST & Fashion MNIST

(b) Fruits

Figure 4: Example inputs and ProtoAttend prototypes for (a) MNIST (with sparsemax), Fashion-MNIST dataset (with sparsemax and sparsity regularization) and (b) Fruits (with sparsemax and sparsity regularization). For MNIST & Fashion-MNIST, prototypes typically consist of discriminative features such as the straight line shape for the digit 1, and the long heels and strips for the sandal. For Fruits, prototypes often correspond to the same fruit captured from a very similar angle.

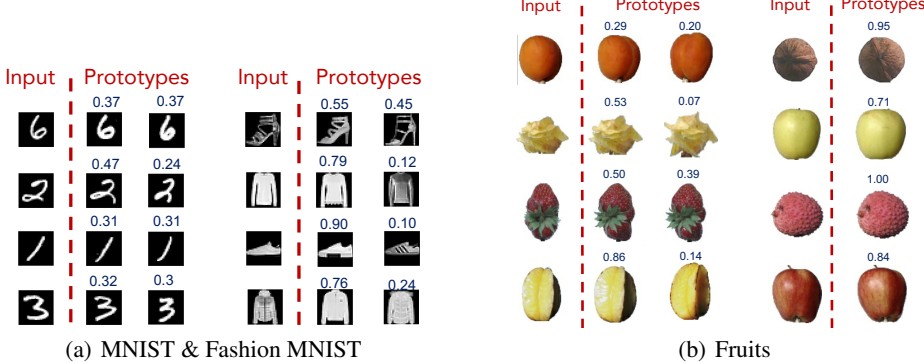

Figure 5: Example inputs and ProtoAttend prototypes for DBPedia (with sparsemax). While classifying the inputs as athlete, prototypes have very similar sentence structure, words and concepts.

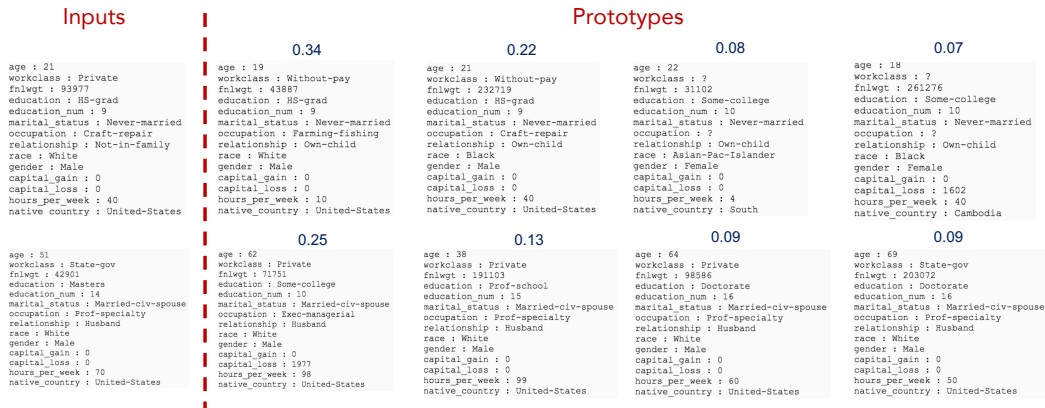

Figure 6: Example inputs and ProtoAttend prototypes for Adult Census Income (with sparsemax and sparsity regularization). For the first example, all prototypes have similar age, two share similar education level and one has the same occupation. For the second example, three prototypes have the same occupation, all work more than 40 hours/week, and three have postgraduate education.

Figs. 4, 5 and 6 exemplify prototypes for image, text and tabular data. In general, perceptually-similar samples are chosen as the prototypes with the largest weights. We also compare the relevant samples found by ProtoAttend with the methods of representer point selection (Yeh et al., 2018) and influence functions (Koh & Liang, 2017) (see Supplementary Material for details) on Animals with Attributes dataset. As shown in Fig. 7, our method finds qualitatively more relevant samples. This case also exemplifies the potential of our method for integration into pre-trained models by addition of simple layers for key, query and value generation.

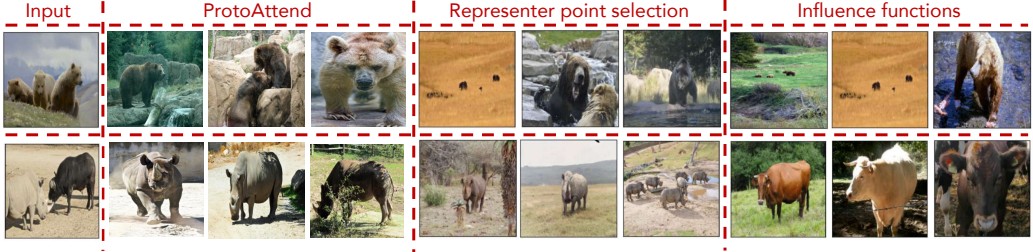

Figure 7: Samples found by ProtoAttend vs. representer point selection (Yeh et al., 2018) and influence function (Koh & Liang, 2017) for the two examples from (Yeh et al., 2018) on Animals with Attributes dataset. See Supplementary Material for more examples.

## 4.3 ROBUSTNESS TO LABEL NOISE

Table 3: Label noise ratio vs. accuracy for baseline encoder, dropout method (Arpit et al., 2017) (optimizing the keep probability) and ProtoAttend with sparsemax attention and sparsity regularization for CIFAR-10.

| Noise level | Test accuracy % | | |
|:---:|:---:|:---:|:---:|
| | Baseline | Dropout | ProtoAttend |
| 0.8 | 57.02 | 56.76 | **60.50** |
| 0.6 | 71.27 | 72.15 | **74.67** |
| 0.4 | 77.47 | 78.99 | **80.04** |

As prototypical learning with sparsemax attention aims to extract decision-making information from a small subset of training samples, it can be used to improve performance when the training dataset

contains noisy labels (see Table 3). The optimal value[11] of $\lambda_{sparse}$ increases with higher noisy label ratios, underlining the increasing importance of sparse learning.

## 4.4 CONFIDENCE-CONTROLLED PREDICTION

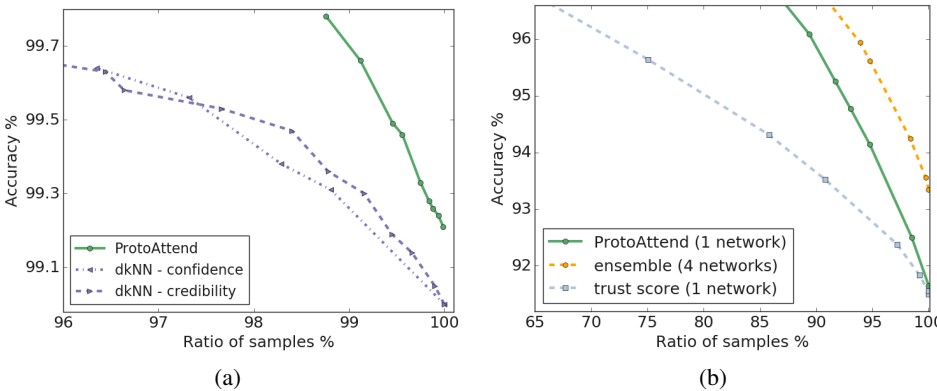

Figure 8: Confidence-controlled prediction. (a) Accuracy vs. ratio of samples for MNIST. We compare dkNN (Papernot & McDaniel, 2018) and prototypical learning (with softmax attention and $\lambda_{conf}$=0.1) using the same network architecture from (Papernot & McDaniel, 2018) without augmentation. (b) Accuracy vs. ratio of samples for CIFAR-10. We compare prototypical learning (with softmax attention and $\lambda_{conf}$=0.1) with trust score (Jiang et al., 2018) and deep ensemble (Lakshminarayanan et al., 2017) methods for the same baseline encoder network architecture.

By varying the threshold for the confidence metric, a trade-off can be obtained for what ratio of the test samples that the model makes a prediction for vs. the overall accuracy it obtains on the samples above that threshold.[12] Figs. 8(a) and 8(b) demonstrate this trade-off and compare it to alternative methods. The sharper slope of the plots show that our method is superior to dkNN (Papernot & McDaniel, 2018) and trust score (Jiang et al., 2018), the methods based on quantifying the mismatch with nearest-neighbor samples, in terms of finding related samples. Although the baseline accuracy is higher with 4 ensemble networks obtained via deep ensemble (Lakshminarayanan et al., 2017), our method utilizes a single network and the additional accuracy gains by refraining from uncertain predictions is similar to our approach as shown by the similar slopes of the curves.

Overall, the baseline accuracy can be significantly improved by making less predictions. Compared to the state of the art models, our canonical method with simple and small models shows similar accuracy by making slightly fewer predictions – e.g. for MNIST, (Wan et al., 2013) achieves 0.21% error rate, that is obtained by our method refraining from only 0.45% of predictions using ResNet-32 and for DBpedia, (Sachan & Petuum, 2018) achieves 0.91% error, that is obtained by our method refraining from 3% of predictions using 9-layer VDCNN. In general, the smaller the number of prototypes, the smaller the trade-off space. Thus, softmax attention (which normally results in more prototypes) is better suited for confidence-controlled prediction compared to sparsemax (see Supplementary Material for more comparisons).

## 4.5 OUT-OF-DISTRIBUTION SAMPLES

Well-calibrated confidence scores at inference can be used to detect deviations from the training dataset. As the test distribution deviates from the training distribution, prototype weights tend to mismatch more and yield lower confidence scores. Fig. 9 (a) shows the ratio of samples above a certain confidence level as the test dataset deviates. Rotations deviate the distribution of test images from the training images, and cause significant degradation in confidence scores, as well as the overall

---

[11]For a fair comparison, we re-optimize the learning rate parameters on a separate validation set.

[12]Note that this trade-off is often more meaningful to consider rather than the metrics based on the actual value of confidence score itself, as methods may differ in how they define the confidence metric, and thus yield very different ranges and distributions for it.

accuracy. On the other hand, using test image from a different dataset, degrade them even further. Next, Fig. 9 (b) shows quantification of out-of-distribution detection with prototypical learning, using the method from (Hendrycks & Gimpel, 2016). ProtoAttend yields an AUC of 0.838, being on par with the-state of the art approaches (Hendrycks et al.).

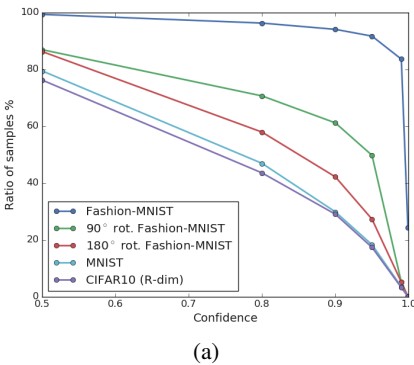
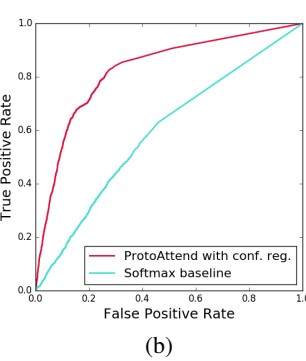

(a)  (b)

Figure 9: Out-of-distribution detection. (a) Ratio of samples above the confidence level for prototypical learning with softmax attention, trained with Fashion-MNIST, and tested on the shown datasets. E.g. if we assess the ratio of samples above confidence 0.9, it is far more likely that those samples come from the same distribution with the training dataset. (b) ROC curve for in-distribution vs. out-of-distribution detection, using CIFAR-10 as in-distribution and SVHN as out-of-distribution, computed using the method from (Hendrycks & Gimpel, 2016) and compared to the proposed baseline in (Hendrycks & Gimpel, 2016). Softmax attention and confidence regularization ($\lambda_{conf} = 0.1$) are used.

## 5  COMPUTATIONAL COST

ProtoAttend requires only a very small increase in the number of learning parameters (merely two extra small matrices for the fully-connected layers to obtain queries and keys). However, it does require a longer training time and has higher memory requirements to process the candidate database. At inference, keys and values for the candidate database can be computed only once and integrated into the model. Thus, the overhead merely becomes the computation of attention outputs (e.g. for CIFAR-10 model, the attention overhead at inference is less than 0.6 MFLOPs, orders of magnitude lower than the computational complexity of a ResNet model). During training on the other hand, both forward and backward propagation steps for the encoder need to be computed for all candidate samples and the total time is higher (e.g. 4.45 times slower to train until convergence for CIFAR-10 compared to the conventional supervised learning). The size of the candidate database is limited by the memory of the processor, so in practice we sample different candidate databases randomly from the training dataset at each iteration. For faster training, data and model parallelism approaches are straightforward to implement – e.g., different processors can focus on different samples, or they can focus on different parts of the convolution or inner product operations. Further computationally-efficient approaches may involve less frequent updates for candidate queries and values.

## 6  CONCLUSIONS

We propose an attention-based prototypical learning method, ProtoAttend, and demonstrate its usefulness for a wide range of problems on image, text and tabular data. By adding a relational attention mechanism to an encoder, prototypical learning enables novel capabilities. With sparsemax attention, it can base the learning on a few relevant samples that can be returned at inference for interpretability, and can also improve robustness to label noise. With softmax attention, it enables confidence-controlled prediction that can outperform state of the art results with simple architectures by simply making slightly fewer predictions, as well as enables detecting deviations from the training data. All these capabilities are achieved without sacrificing overall accuracy of the base model.

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

## A  PSEUDO CODE FOR TRAINING

---
**Algorithm 1** Pseudo-code of ProtoAttend training
---

1: **Inputs:** Training dataset $\mathcal{T}$, encoder model $h(\mathbf{x}; \theta)$, classifier model $h(\mathbf{v}; \phi)$, normalization function $n$, input batch size $B$, candidate batch size $D$, attention dimension $d_{att}$, $\alpha$ values to be used for loss: (0, 0.5, 1), task-specific loss function $L$, ADAM learning rate $r$, and exponential decay rate parameters $\beta_1$ and $\beta_2$

2: **Initialize** Trainable encoder parameters $\theta$ and classifier layer parameters $\phi$

3: **while** until convergence **do**

4:     Sample a mini-batch from the training dataset for the inputs: $(\mathbf{x}_i, y_i)_{i=1}^{B} \sim \mathcal{T}$

5:     Sample a mini-batch from the training dataset for the prototypes: $(\mathbf{x}_j^{(c)}, y_j^{(c)})_{j=1}^{D} \sim \mathcal{T}$

6:     **for** $i = 1, ..., B$ **do**

7:         Obtain queries and values for the input:

$$\mathbf{Q_i}, \mathbf{V_i} \leftarrow h(\mathbf{x}; \theta)$$

8:     **for** $j = 1, ..., D$ **do**

9:         Obtain keys and values for the prototypes:

$$\mathbf{K_j^{(c)}}, \mathbf{V_j^{(c)}} \leftarrow h(\mathbf{x}^{(c)}; \theta)$$

10:     **for** $i = 1, ..., B$ **do**

11:         **for** $j = 1, ..., D$ **do**

12:             Estimate the relational attention coefficients:

$$p_{i,j} \leftarrow n\left(\mathbf{K_j^{(c)}}\mathbf{Q_i}^T / \sqrt{d_{att}}\right)$$

13:     Obtain the predictions

14:     **for** $i = 1, ..., B$ **do**

$$\hat{y}_i(\alpha = 0) \leftarrow g\left(\mathbf{v_i}; \phi\right)$$

$$\hat{y}_i(\alpha = 0.5) \leftarrow g\left(0.5\mathbf{v_i} + 0.5\sum\nolimits_{j=1}^{D} p_{i,j}\mathbf{v_j^{(c)}}; \phi\right)$$

$$\hat{y}_i(\alpha = 1) \leftarrow g\left(\sum\nolimits_{j=1}^{D} p_{i,j}\mathbf{v_j^{(c)}}; \phi\right)$$

15:     Estimate the total loss function

$$L_{batch} \leftarrow 1/B \cdot \sum\nolimits_{i=1}^{B} L\left(y_i, \hat{y}_i(0)\right) + L\left(y_i, \hat{y}_i(1)\right) + L\left(y_i, \hat{y}_i(0.5)\right)$$

16:     Update the encoder model and the classifier layer

$$\phi \leftarrow \phi - \text{ADAM}(\nabla_\phi L_{batch}, r, \beta_1, \beta_2)$$

$$\theta \leftarrow \theta - \text{ADAM}(\nabla_\theta L_{batch}, r, \beta_{1,2})$$

---

## B  RELATION TO INFLUENCE FUNCTIONS

Here we clarify the relationship of our work to influence functions from a theoretical perspective. Influence functions quantify how a model's predictions would change if we did not have a particular training point. For the purpose of sample-based explainability, (Koh & Liang, 2017) proposes that the relation between an input sample $\mathbf{x_i}$ and the candidate samples[13] $\mathbf{x_j^{(c)}}$ can be obtained by quantifying

---

[13]All training samples are used as candidate samples in (Koh & Liang, 2017).

the influence of upweighting $(\mathbf{x_j^{(c)}}, y_j^{(c)})$ on the loss at a query point $(\mathbf{x_i}, y_i)$:

$$\mathcal{I}_{i,j} = -\nabla_{(\theta,\phi)} L(y_j^{(c)})^T (\mathbf{H}_{(\hat{\theta},\hat{\phi})}^{-1})^T \nabla_{(\theta,\phi)} L(y_i), \tag{4}$$

where $\mathbf{H}_{(\hat{\theta},\hat{\phi})}$ is the Hessian and is positive definite by assumption. Let's consider the singular value decomposition $(\mathbf{H}_{(\hat{\theta},\hat{\phi})}^{-1}) = \mathbf{\Xi} \cdot \mathbf{\Sigma} \cdot \mathbf{\Psi}^T$ and also define the function $k(\mathbf{x}, y) = \nabla_{(\theta,\phi)} L(y)$. Then, Eq. 4 can be written as:

$$\mathcal{I}_{i,j} = (\mathbf{\Psi}^T \cdot k(\mathbf{x_j^{(c)}}, y_j^{(c)}))^T \cdot (-\mathbf{\Sigma} \cdot \mathbf{\Xi}^T) \cdot k(\mathbf{x_i}, y_i), \tag{5}$$

We can observe that $\mathcal{I}_{i,j}$ is in the form of an inner product between two functions applied on $(\mathbf{x_i}, y_i)$ and $(\mathbf{x_j^{(c)}}, y_j^{(c)})$. These two functions are composed of a shared (and potentially complex) function, followed by a linear mapping with non-shared parameters. This expression is indeed in a similar form with the argument of the normalization function for attention in Eq. 1, where the queries and keys are obtained by a shared encoder except the last layer. The only notable difference is that ProtoAttend encoder functions merely input $\mathbf{x_i}$ and $\mathbf{x_j^{(c)}}$, not the ground truth labels. Instead of relying on ground truth labels or complex Hessian estimations, ProtoAttend infers the encoded representations for the queries and keys directly in a feedforward way, by learning from the entire training dataset. Note that ProtoAttend does not use a separate encoder for values, and obtains a high performance by sharing the vast majority of the parameters while obtaining the keys, queries and values.

In (Koh & Liang, 2017), Influence Functions are also related to nearest neighbor search-based relevant point determination approaches, for sample-based explainability. When Euclidean space is considered for distances, with the assumption that all points have the same norm, the inner product between the representations correspond to their similarity. This scenario is the special case of ProtoAttend when we use the same representation for keys, queries and values, and when we train with only $\alpha = 0$ loss term although we would use $p_{i,j}$ for similarity determination. As studied in (Koh & Liang, 2017), nearest neighbor-based methods are far less accurate in capturing the effect of model training, compared to Influence Functions. Our empirical results in Figs. 7 and 13 show superior performance of ProtoAttend compared to Influence Functions in finding perceptually more similar samples.

Overall, unlike Influence Functions, ProtoAttend modifies the model training for the desired goals, that fundamentally yields more degrees of freedom to optimize while achieving superior prototype learning quality effectively.

## C  TRAINING DETAILS

Different candidate databases are sampled randomly from the training dataset at each iteration. Training database size is chosen to fit the model to the memory of a single GPU. $D$ at inference is chosen sufficiently large to obtain high accuracy. Table 4 shows the database size $D$ for the datasets used in the experiments. The size of the prototype candidate database should be sufficiently large such that the model can attend to reasonable prototypes with high coefficients (separately for each input). With appropriate sparsity mechanisms, we normally only end up with a few prototypes with large coefficients. Indeed, most of the coefficients would be zero with sparsemax activation and sparsity regularization.

Table 4: Datasets and database size $D$.

| Dataset | Encoder | Database size $D$ | |
|---|---|---|---|
| | | Training | Inference |
| MNIST | ResNet | 1024 | 32768 |
| Fashion-MNIST | ResNet | 1024 | 32768 |
| CIFAR-10 | ResNet | 1024 | 32768 |
| Fruits | ResNet | 256 | 4096 |
| ISIC Melanoma | ResNet | 256 | 4096 |
| DBPedia | VDCNN | 512 | 4096 |
| Census Income | LSTM | 4096 | 15360 |

### C.1 IMAGE DATA

### C.1.1 MNIST DATASET

We apply random cropping after padding each side by 2 pixels and per image standardization. The base encoder uses a standard 32 layer ResNet architecture. The number of filters is initially 16 and doubled every 5 blocks. In each block, two $3 \times 3$ convolutional layers are used to transform the input, and the transformed output is added to the input after a $1 \times 1$ convolution. $4\times$ downsampling is applied by choosing the stride as 2 after $5^{th}$ and $10^{th}$ blocks. Each convolution is followed by batch normalization and ReLU nonlinearity. After the last convolution, $7 \times 7$ average pooling is applied. The output is followed by a fully-connected layer of 256 units and ReLU nonlinearity, followed by layer normalization (Lei Ba et al., 2016). Keys and queries are mapped from the output using a fully-connected layer followed by ReLU nonlinearity, where the attention size is $d_{att}$=16. Values are mapped from the output using a fully-connected layer of $d_{out}$=64 units and ReLU nonlinearity, followed by layer normalization. For the baseline encoder, the initial learning rate is chosen as 0.002 and exponential decay is applied with a rate of 0.9 applied every 6k iterations. The model is trained for 84k iterations. For prototypical learning model with softmax attention, the initial learning rate is chosen as 0.002 and exponential decay is applied with a rate of 0.8 applied every 8k iterations. The model is trained for 228k iterations. For prototypical learning model with sparsemax attention, the initial learning rate is chosen as 0.001 and exponential decay is applied with a rate of 0.93 applied every 6k iterations. The model is trained for 228k iterations. All models use a batch size of 128 and gradient clipping above 20.

### C.1.2 FASHION-MNIST DATASET

We apply random cropping after padding each side by 2 pixels, random horizontal flipping, and per image standardization. The base encoder uses a standard 32 layer ResNet architecture, similar to our MNIST experiments. For the baseline encoder, the initial learning rate is chosen as 0.0015 and exponential decay is applied with a rate of 0.9 applied every 10k iterations. The model is trained for 332k iterations. For prototypical learning with softmax attention, the initial learning rate is chosen as 0.0007 and exponential decay is applied with a rate of 0.92 applied every 8k iterations. The model is trained for 450k iterations. For prototypical learning with sparsemax attention, the initial learning rate is chosen as 0.001 and exponential decay is applied with a rate of 0.9 applied every 8k iterations. The model is trained for 392k iterations. For prototypical learning with sparsemax attention and sparsity regularization (with $\lambda_{sparse} = 0.0003$), the initial learning rate is chosen as 0.001 and exponential decay is applied with a rate of 0.94 applied every 8k iterations. $\lambda_{conf} = 0.1$ is chosen when confidence regularization is applied. The model is trained for 440k iterations. All models use a batch size of 128 and gradient clipping above 20.

### C.1.3 CIFAR-10 DATASET

We apply random cropping after padding each side by 3 pixels, random horizontal flipping, random vertical flipping and per image standardization. The base encoder uses a standard 50 layer ResNet architecture. The number of filters is initially 16 and doubled every 8 blocks. In each block, two $3 \times 3$ convolutional layers are used to transform the input, and the transformed output is added to the input after a $1 \times 1$ convolution. $4\times$ downsampling is applied by choosing the stride as 2 after $8^{th}$ and $16^{th}$ blocks. Each convolution is followed by batch normalization and the ReLU nonlinearity. After the last convolution, $8 \times 8$ average pooling is applied. The output is followed by a fully-connected layer of 256 units and the ReLU nonlinearity, followed by layer normalization (Lei Ba et al., 2016). Keys and queries are mapped from the output using a fully-connected layer followed by the ReLU nonlinearity, where the attention size is $d_{att}$=16. Values are mapped from the output using a fully-connected layer of $d_{out}$=128 units and the ReLU nonlinearity, followed by layer normalization. For the baseline encoder, the initial learning rate is chosen as 0.002 and exponential decay is applied with a rate of 0.95 applied every 10k iterations. The model is trained for 940k iterations. For prototypical learning with softmax attention, the initial learning rate is chosen as 0.0035 and exponential decay is applied with a rate of 0.95 applied every 10k iterations. The model is trained for 625k iterations. For prototypical learning with sparsemax attention, the initial learning rate is chosen as 0.0015 and exponential decay is applied with a rate of 0.95 applied every 10k iterations. The model is trained for 905k iterations. For prototypical learning with sparsemax attention and sparsity regularization

(with $\lambda_{sparse} = 0.00008$), the initial learning rate is chosen as 0.0015 and exponential decay is applied with a rate of 0.95 applied every 12k iterations. $\lambda_{conf} = 0.1$ is chosen when confidence regularization is applied. The model is trained for 450k iterations. All models use a batch size of 128 and gradient clipping above 20.

**CIFAR-10 experiments with noisy labels.** For CIFAR-10 experiments with noisy labels for the base encoder we only optimize the learning parameters. Noisy labels are sampled uniformly from the set of labels excluding the correct one. The baseline model with noisy label ratio of 0.8 uses an initial learning rate of 0.001, decayed with a rate of 0.92 every 6k iterations, and is trained for 15k iterations. For the dropout approach, dropout with a rate of 0.1 is applied, and the model uses an initial learning rate of 0.002, decayed with a rate of 0.85 every 8k iterations, and is trained for 24k iterations. The baseline model with noisy label ratio of 0.6 uses an initial learning rate of 0.002, decayed with a rate of 0.92 every 6k iterations, and is trained for 12k iterations. For the dropout approach, dropout with a rate of 0.3 is applied, and the model uses an initial learning rate of 0.002, decayed with a rate of 0.92 every 8k iterations, and is trained for 18k iterations. The baseline model with noisy label ratio of 0.4 uses an initial learning rate of 0.002, decayed with a rate of 0.92 every 6k iterations, and is trained for 15k iterations. For the dropout approach, dropout with a rate of 0.5 is applied, and the model uses an initial learning rate of 0.002, decayed with a rate of 0.92 every 6k iterations, and is trained for 18k iterations. For experiments for the prototypical learning model with sparsemax attention, we optimize the learning parameters and $\lambda_{sparse}$. For the model with noisy label ratio of 0.8, $\lambda_{sparse} = 0.0015$, initial learning rate is chosen as 0.0006 and exponential decay is applied with a rate of 0.95 applied every 8k iterations. The model is trained for 108k iterations. For the model with noisy label ratio of 0.6, $\lambda_{sparse} = 0.0005$, initial learning rate is chosen as 0.001 and exponential decay is applied with a rate of 0.9 applied every 8k iterations. The model is trained for 92k iterations. For the model with noisy label ratio of 0.4, $\lambda_{sparse} = 0.0003$, initial learning rate is chosen as 0.001 and exponential decay is applied with a rate of 0.9 applied every 6k iterations. The model is trained for 122k iterations.

### C.1.4 FRUITS DATASET

We apply random cropping after padding each side by 5 pixels, random horizontal flipping, random vertical flipping and per image standardization. In the encoder, first, a downsampling with a convolutional layer is applied with a stride of 2, and using 16 filters, followed by a downsampling with max-pooling with a stride of 2. After obtaining the $25 \times 25$ inputs, a standard 32 layer ResNet architecture (similar to MNIST) is used, followed by a fully-connected layer of 128 units and the ReLU nonlinearity, followed by layer normalization (Lei Ba et al., 2016). Keys and queries are mapped from the output using a fully-connected layer followed by the ReLU nonlinearity, where the attention size is $d_{att}$=16. Values are mapped from the output using a fully-connected layer of $d_{out}$=64 units and the ReLU nonlinearity, followed by layer normalization. W eight decay with a factor of 0.0001 is applied for the convolutional filters. The model uses a batch size of 128 and gradient clipping above 20.

### C.1.5 ISIC MELANOMA DATASET

The ISIC Melanoma dataset is formed from the ISIC Archive (ISIC, 2016) that contains over 13k dermoscopic images collected from leading clinical centers internationally and acquired from a variety of devices within each center. The dataset consists of skin images with labels denoting whether they contain melanoma or are benign. We construct the training and validation dataset using 15122 images (13511 benign and 1611 melanoma cases), and the evaluation dataset using 3203 images (2867 benign and 336 melanoma). While training, benign cases are undersampled in each batch to have 0.6 ratio including candidate database sets at training and inference. All images are resized to $128 \times 128$ pixels. We apply random cropping after padding each side by 8 pixels, random horizontal flipping, random vertical flipping and per image standardization. In the encoder, first, a downsampling with a convolutional layer is applied with a stride of 2, and using 16 filters, followed by a downsampling with max-pooling with a stride of 2. After obtaining the $32 \times 32$ inputs, the base encoder uses a standard 50 layer ResNet architecture (similar to CIFAR10), followed by a fully-connected layer of 128 units and the ReLU nonlinearity, followed by layer normalization (Lei Ba et al., 2016). Keys and queries are mapped from the output using a fully-connected layer followed by the ReLU nonlinearity, where the attention size is $d_{att}$=16. Values are mapped from the

output using a fully-connected layer of $d_{out}$=64 units and the ReLU nonlinearity, followed by layer normalization. For the baseline encoder, the initial learning rate is chosen as 0.002 and exponential decay is applied with a rate of 0.9 applied every 3k iterations. The model is trained for 220k iterations. For prototypical learning with softmax attention, the initial learning rate is chosen as 0.0006 and exponential decay is applied with a rate of 0.9 applied every 3k iterations. The model is trained for 147k iterations. For prototypical learning with sparsemax attention, the initial learning rate is chosen as 0.0006 and exponential decay is applied with a rate of 0.9 applied every 4k iterations. The model is trained for 166k iterations. All models use a batch size of 128 and gradient clipping above 20.

### C.1.6 ANIMALS WITH ATTRIBUTES DATASET

We train ProtoAttend with sparsemax attention using the features from a pre-trained ResNet-50 as provided in (Yeh et al., 2018). To map the pre-trained features, we simply insert a single fully-connected layer with 256 units with ReLU nonlinearity and layer normalization, followed by the individual fully-connected layers of keys, queries and values (16, 16 and 64 units respectively with ReLU nonlinearity). Sparsity regularization is applied with $\lambda_{sparse} = 0.000001$. We train the model for 70k iterations. The initial learning rate is chosen as 0.0006 and exponential decay is applied with a rate of 0.8 applied every 10k iterations. A classification accuracy above 91% is obtained for the test set.

## C.2 TEXT DATA

### C.2.1 DBPEDIA DATASET

There are 14 output classes: Company, Educational Institution, Artist, Athlete, Office Holder, Mean Of Transportation, Building, Natural Place, Village, Animal, Plant, Album, Film, Written Work. As the input, 16-dimensional trainable embeddings are mapped from the dictionary of 69 raw characters (Conneau et al., 2016). The maximum length is set to 448 and longer inputs are truncated while the shorter inputs are padded. The input embeddings are first transformed with a 1-D convolutional block consisting 64 filters with kernel width of 3 and stride of 2. Then, 8 convolution blocks as in (Conneau et al., 2016) are applied, with 64, 64, 128, 128, 256, 256, 512 and 512 filters respectively. All use the kernel width of 3, and after each two layers, max pooling is applied with kernel width of 3 and a stride of 2. All convolutions are followed by batch normalization and the ReLU nonlinearity. Convolutional filters use weight normalization with parameter 0.00001. The last convolution block is followed by k-max pooling with $k$=8 (Conneau et al., 2016). Finally, we apply two fully-connected layers with 1024 hidden units. In contrast to (Conneau et al., 2016), we also use layer normalization (Lei Ba et al., 2016) after fully-connected layers as we observe this leads to more stable training behavior. Keys and queries are mapped from the output using a fully-connected layer followed by the ReLU nonlinearity, where the attention size is $d_{att}$=16. Values are mapped from the output using a fully-connected layer of $d_{out}$=64 units and the ReLU nonlinearity, followed by layer normalization. For the baseline encoder, initial learning rate is chosen as 0.0008 and exponential decay is applied with a rate of 0.9 applied every 8k iterations. The model is trained for 212k iterations. For prototypical learning model with softmax attention, the initial learning rate is chosen as 0.0008 and exponential decay is applied with a rate of 0.9 applied every 8k iterations. The model is trained for 146k iterations. For prototypical learning model with sparsemax attention, the initial learning rate is chosen as 0.0005 and exponential decay is applied with a rate of 0.82 applied every 8k iterations. The model is trained for 270k iterations. All models use a batch size of 128 and gradient clipping above 20. We do not apply any data augmentation.

## C.3 TABULAR DATA

### C.3.1 ADULT CENSUS INCOME

There are two output classes: whether or not the annual income is above $50k. Categorical categories such as the 'marital-status' are mapped to multi-hot representations. Continuous variables are used after a fixed normalization transformation. For 'age', the transformation first subtracts 50 and then divides by 30. For 'fnlwgt', the transformation first takes the log, and then subtracts 9, and then divides by 3. For 'education-num', the transformation first subtracts 6 and then divides by 6. For 'hours-per-week', the transformation first subtracts 50 and then divides by 50. For 'capital-gain'

and 'capital-loss', the normalization takes the log, and then subtracts 5, and then divides by 5. The concatenated features are then mapped to a 64 dimensional vector using a fully-connected layer, followed by the ReLU nonlinearity. The base encoder uses an LSTM architecture, with 4 timesteps. At each timestep, 64-dimensional inputs are applied after a dropout with rate 0.5. The output of the last timestep is used after applying a dropout with rate 0.5. Keys and queries are mapped from this output using a fully-connected layer followed by the ReLU nonlinearity, where the attention size is $d_{att}$=16. Values are mapped from the output using a fully-connected layer of $d_{out}$=16 units and the ReLU nonlinearity, followed by layer normalization. For the baseline encoder, the initial learning rate is chosen as 0.002 and exponential decay is applied with a rate of 0.9 applied every 2k iterations. The model is trained for 4.5k iterations. For the models with attention in prototypical learning framework, the initial learning rate is chosen as 0.0005 and exponential decay is applied with a rate of 0.92 applied every 2k iterations. The softmax attention model is trained for 13.5k iterations and the sparsemax attention model is trained for 11.5k iterations. For the model with sparsity regularization, the initial learning rate is 0.003 and exponential decay is applied with a rate of 0.7 applied every 2k iterations, and the model is trained for 7k iterations. All models use a batch size of 128 and gradient clipping above 20. We do not apply any data augmentation.

## D  ADDITIONAL PROTOTYPE EXAMPLES

Fig. 10 exemplify prototypes for CIFAR-10. For most cases, we observe the similarity of discriminative features between inputs and prototypes. For example, the body figures of birds, the shape of tires, the face patterns of dogs, the body figures of frogs, the appearance of the background sky for planes, are among the features apparent in examples.

Fig. 11 shows additional prototype examples for DBPedia dataset. Prototypes have very similar sentence structure, words and concepts, while categorizing the sentences into ontologies.

Fig. 12 shows example prototypes for ISIC Melanoma. In some cases, we observe the commonalities between input and prototypes that distinguish melanoma cases such as the non-circular geometry or irregularly-notched borders (Jerant et al., 2000). Compared to other datasets, ISIC Melonama dataset yields lower interpretable prototype quality on average. We hypothesize this to be due to the perceptual difficulty of the problem as well as the insufficient encoder performance shown by the lower classification accuracy (despite the acceptable AUC).

Fig. 13 shows more comparison examples for prototypical learning framework with sparsemax attention vs. representer point selection (Yeh et al., 2018) on Animals with Attributes dataset. For some cases, including chimpanzee, zebra, dalmatian and tiger, ProtoAttend yields perceptually very similar samples. The similarity of the chimpanzee body form and the background, zebra patterns, dalmatian pattern on the grass, and tiger pattern and head pose, are prominent. Representer point selection fails to capture such similarity features as effectively. On the other hand, for bat, otter and wolf, the results are somewhat less satisfying. The wing part of the bat, multiple count of the otters with the background, and the color and furry head of the wolf seem to be captured, but with less apparent similarity than some other possible samples from the dataset. Representer point selection method also cannot be claimed to be successful in these cases. Lastly, for leopard, ProtoAttend only yields one non-zero prototype (which is indeed statistically rare given the model and sparsity choices). The pattern of the leopard image seems relevant, but it is also not fully satisfying to observe a single prototype that is not perceptually more similar. All of the test examples in Fig. 13 are classified correctly with our framework and all of the shown prototypes are also from the correct classes.

## E  COMPARISON OF CONFIDENCE-CONTROLLED PREDICTION FOR SOFTMAX VS. SPARSEMAX

Figs. 14 and 15 show the accuracy vs. ratio of samples for softmax vs. sparsemax attention without confidence regularization. The baseline accuracy (at 100% prediction ratio) is higher for softmax attention for some datasets, whereas higher for sparsemax for some others. On the other hand, higher number of prototypes yielded by softmax attention results in a wider range for confidence-controlled prediction trade-off.

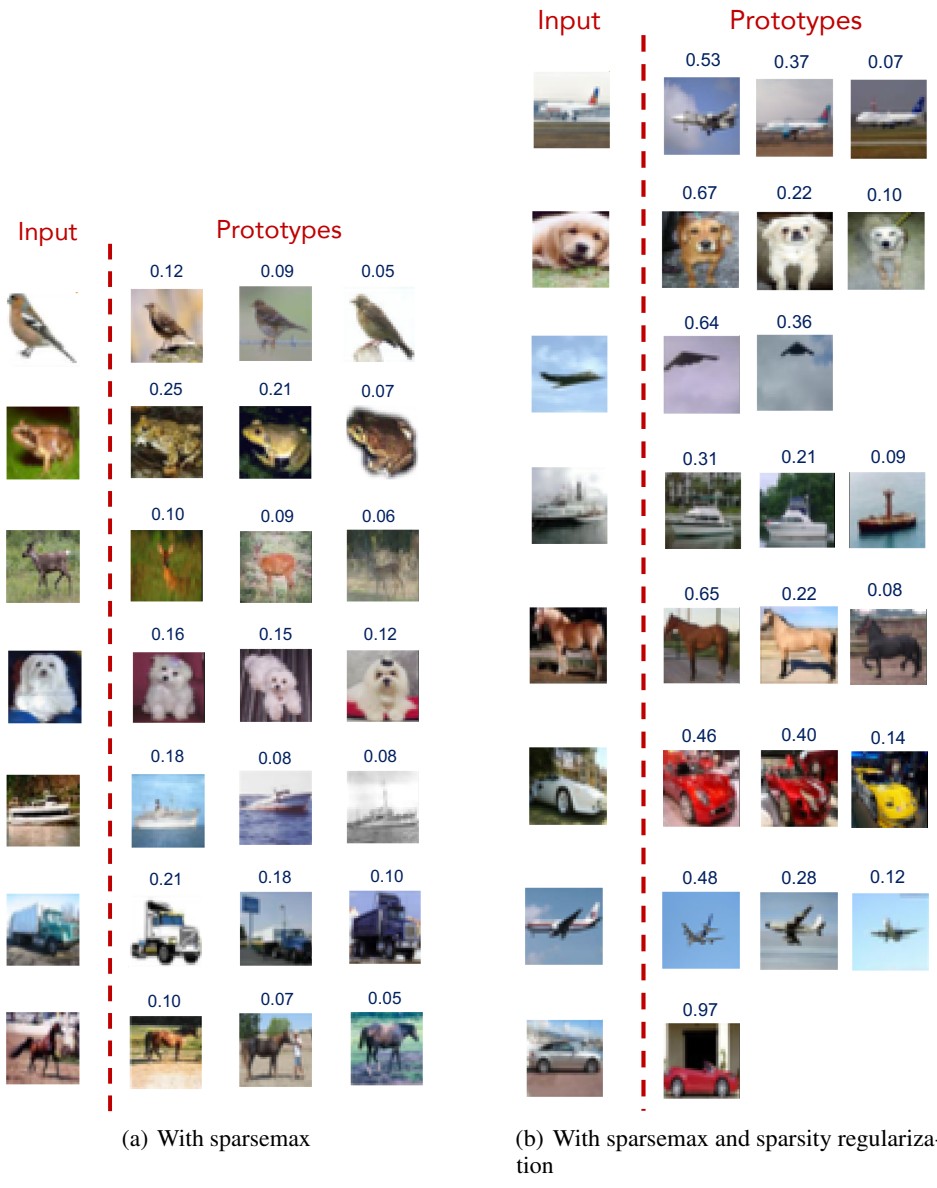

(a) With sparsemax

(b) With sparsemax and sparsity regularization

Figure 10: Example inputs and corresponding prototypes for CIFAR-10.

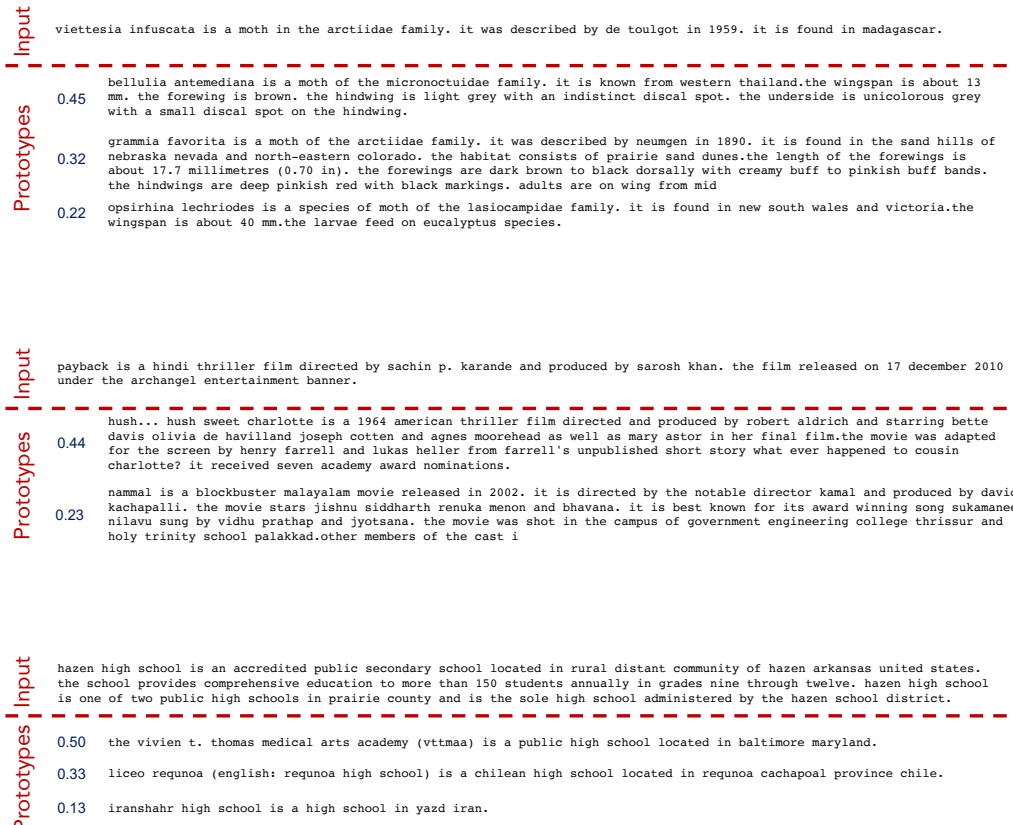

Figure 11: Example inputs and corresponding prototypes for DBPedia (with sparsemax).

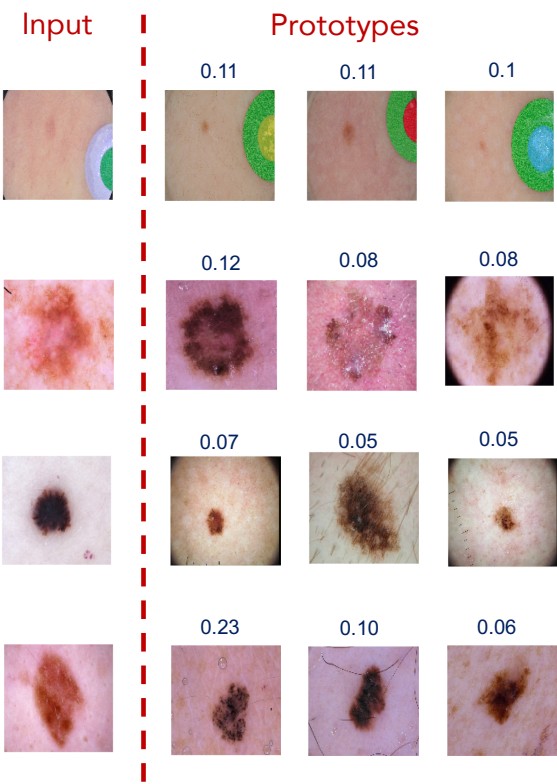

Figure 12: Example inputs and corresponding prototypes for ISIC Melanoma (with sparsemax attention).

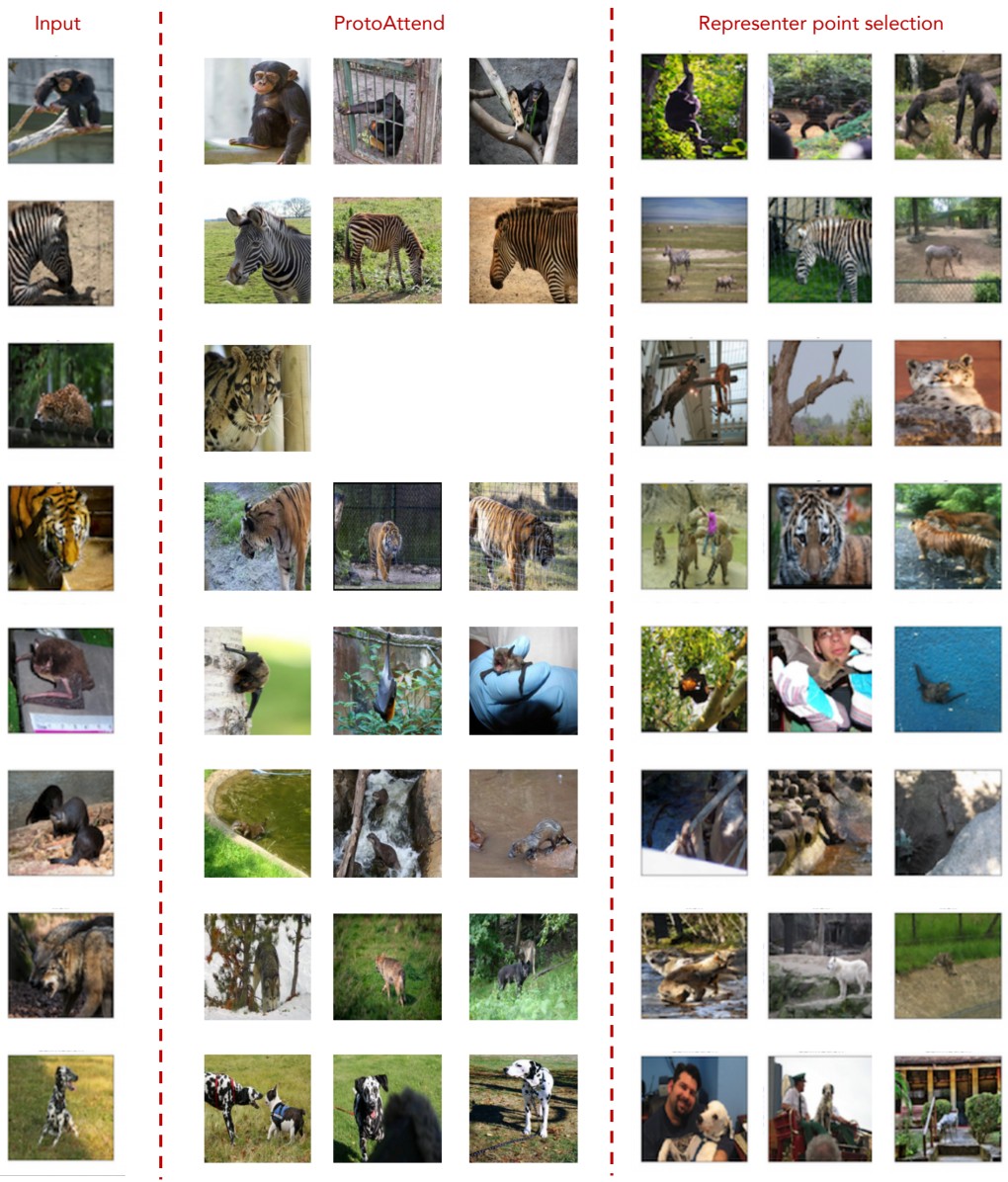

Figure 13: Relevant samples found by ProtoAttend with sparsemax attention vs. representer point selection (Yeh et al., 2018) for the examples from Supplementary Material of (Yeh et al., 2018).

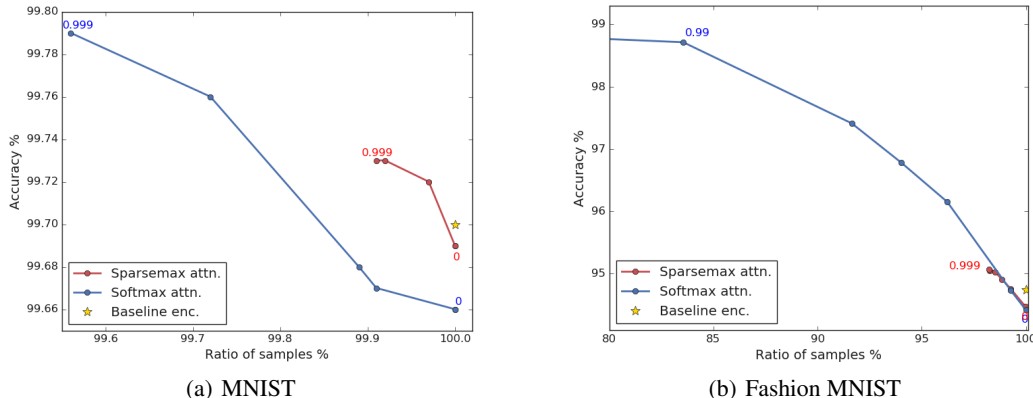

(a) MNIST

(b) Fashion MNIST

Figure 14: Accuracy vs. ratio of samples for (a) MNIST and (b) Fashion MNIST, for confidence levels between 0 and 0.999.

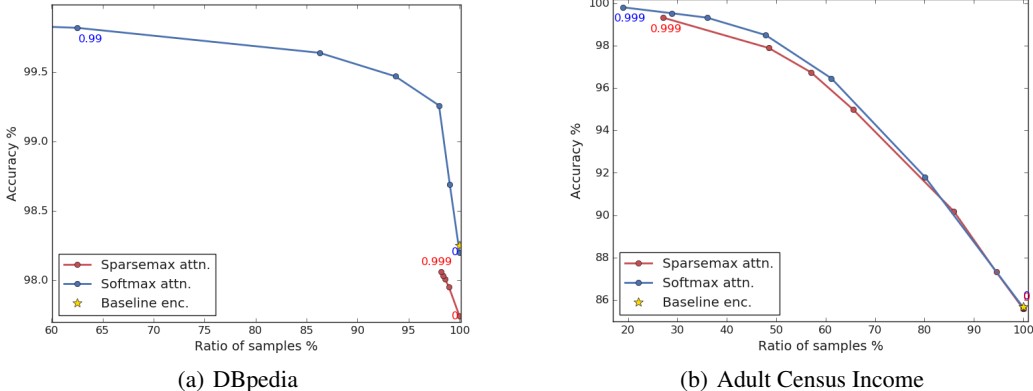

(a) DBpedia

(b) Adult Census Income

Figure 15: Accuracy vs. ratio of samples for (a) DBpedia and (b) Adult Census Income, for confidence levels between 0 and 0.999.

As an impactful case study, we consider melanoma detection problem with ISIC dataset (ISIC, 2016) in Supplementary Material. In medical diagnosis, it is strongly desired to maintain a sufficiently-high prediction performance, potentially by verifying the decisions of an AI system by medical experts in the cases where the AI models are not confident. By refraining from some predictions, as shown in Fig. 16, we demonstrate unprecedentedly high AUC values without using transfer learning or highly-customized models (Haenssle et al., 2018).

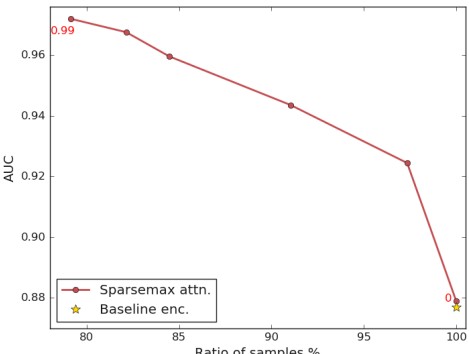

Figure 16: Area-under-curve (AUC) vs. ratio of samples for ISIC Melanoma with softmax attention, for confidence values ranging between 0 and 0.99.

## F   HUMAN USER STUDY ON THE USEFULNESS OF PROTOTYPES

We perform a user study by asking humans how much an extra image helps in explaining the guessed class of the input, after showing what the trained network predicts for that input. We consider the Animals with Attributes dataset (exemplified in Fig. 13). We randomly pick test samples and assess how much showing the top prototype makes a difference. The results in Table 5 shows that ProtoAttend picks

Table 5: Human ratings (mean score and 95% confidence interval) on how much an extra image helps guessing the class of the input.

| Sampling method | Score (out of 5) |
|---|---|
| Top prototype by ProtoAttend | $4.33 \pm 0.09$ |
| Randomly sampled from the predicted class | $3.97 \pm 0.12$ |
| Randomly sampled from any class | $1.33 \pm 0.09$ |

## G   CONTROLLING SPARSITY VIA REGULARIZATION

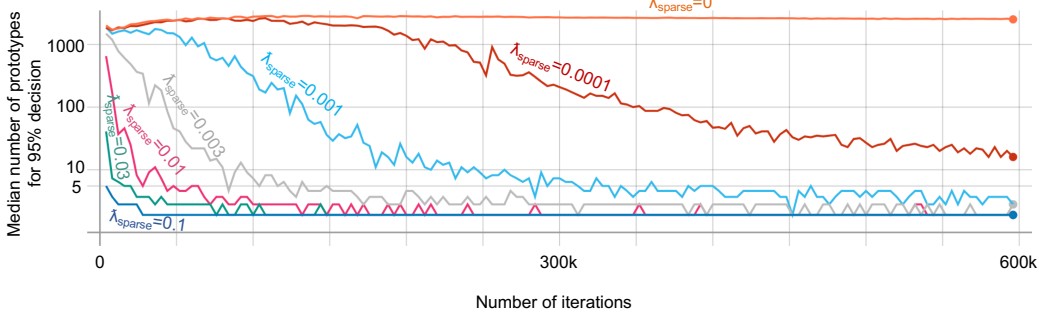

Figure 17: Number of training iterations vs. median number prototypes to explain 95% of the decision (in logarithmic scale), for Fashion-MNIST with softmax attention.

Fig. 17 shows the impact of sparsity regularization coefficient on training. By varying the value of $\lambda_{sparse}$, the number of prototypes can be efficiently controlled. For high values of sparsity regularization coefficient, the model gets stuck at a point where it is forced to make decision from a low number of prototypes before the encoder model is properly learned, hence typically yields considerably lower performance. We also observe sparsity mechanism via sparsemax attention to yield better performance than softmax attention with high sparsity regularization.

## H  PROTOTYPE QUALITY

In general, the following scenarios may yield low prototype quality:

1. Lack of related samples in the candidate database.
2. Perceptual difference between humans and encoders in determining discriminative features.
3. High intra-class variability that makes training difficult.
4. Imperfect encoder that cannot yield fully accurate representations of the input.
5. Insufficiency of relational attention to determine weights from queries and keys.
6. Inefficient decoupling between encoder & attention blocks and the final decision block.

There can be problem-dependent fundamental limitations on (1)-(3), whereas (4)-(6) are raised by choices of models and losses and can be further improved. We leave the quantification of prototype quality using information-theoretic metrics or discriminative neural networks to future work.

## I  UNDERSTANDING MISCLASSIFICATION CASES

One of the benefits of prototypical learning is insights into wrong decision cases. Fig. 18 exemplifies prototypes with wrong labels, that give insights about why the model is confused about a particular input (e.g. due to similarity of the visual patterns). Such insights can be actionable to improve the model performance, such as adding more training samples for the confusing classes or modifying the loss functions.

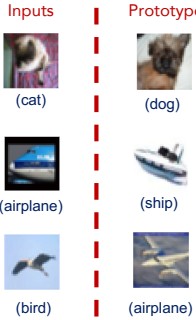

Figure 18: Example prototypes with wrong labels for CIFAR-10.

