# OpenReview forum: "ProtoAttend: Attention-Based Prototypical Learning"
_ICLR.cc/2020/Conference — Reject_

### Official Review · AnonReviewer3 · 2019-10-23
**Official Blind Review #3**

**Rating:** 3

**Review:**

This work proposes an attention-based prototype learning algorithm, which introduces an attention operation to assign different weights to the prototypes. Comprehensive experiments demonstrate that the proposed method is efficient and effective in various tasks.

I have the following comments:
-	The authors did a really good job in empirical studies, verifying the superiority of ProtoAttend.

-	The novelty of the main idea is limited and may provide a limited contribution to the research community.

-	The authors clarified that ProtoAttend is an inherently interpretable algorithm. However, the interpretability is proved by naïve Prototype learning only. A rigorous theoretical proof should be provided to demonstrate its interpretability.

-	I would be appreciated if the authors provide the pseudo-code to show the training procedure of ProtoAttend.

Overall, I think this work is not ready for publishing unless the theoretical property is well understood.


**Experience Assessment:**

I have read many papers in this area.

**Review Assessment: Checking Correctness Of Derivations And Theory:**

I carefully checked the derivations and theory.

**Review Assessment: Checking Correctness Of Experiments:**

I assessed the sensibility of the experiments.

**Review Assessment: Thoroughness In Paper Reading:**

I read the paper at least twice and used my best judgement in assessing the paper.

---

> ### Author Response · Authors · 2019-11-13
> **Response to Review #3**
>
> Thanks for your valuable comments and finding our method efficient and effective!
>
> I have the following comments:
> Q1: The authors did a really good job in empirical studies, verifying the superiority of ProtoAttend.
>
> A1: We really appreciate that you acknowledge our contributions that are demonstrated with strong empirical studies.
>
> Q2: The novelty of the main idea is limited and may provide a limited contribution to the research community.
>
> A2: We clarify the novelty and research community contribution below.
>
> 1) Why is this topic important?
>
> Explainability is clearly an important bottleneck for widespread adoption of AI [1].  For some of the most impactful and highest value applications of AI, such as in healthcare, finance or retail, there are clear use cases for sample-based interpretability [1] [2] [3].  Thus, we believe contributions in this area of sample-based interpretability are of significant importance to AI in general and the research community in particular.
>
> 2) Why is this work novel?
>
> This is the first time sample-based interpretability is integrated in machine learning architecture design.  This is in contrast to previous straightforward post-hoc approaches or simple modifications to the loss function.  To this end ProtoAttend has multiple important and canonical contributions in inherently-interpretable deep neural network design:
> 1) New design principles that, going forward, can guide the design of inherently-interpretable models based on sample-based interpretability.
> 2) A novel attention method for sample-based interpretability that fulfils these principles by selecting input-dependent prototypes based on an attention mechanism between the input and prototype candidates.
> 3) Our design approach is datatype and model-agnostic, meaning it can have a wide range of applications, and even has the ability to be integrated with pre-trained models.
> 4) We demonstrate for the first time how sample-based interpretability can be used to improve other capabilities like confidence estimation, noise-robust learning or out-of-distribution detection.
> 5) We design a method that enables all these benefits via the same architecture and method while maintaining comparable overall accuracy to the base model.
>
> We have modified our Introduction section to reflect these points more clearly and hope that clarifies this question.
>
> [1] “Explainable Artificial Intelligence (XAI): Concepts, Taxonomies, Opportunities and Challenges toward Responsible AI”, by A. B. Arrieta.
> [2] “Efficient Data Representation by Selecting Prototypes with Importance Weights”, by K S. Gurumoorthy et al.
> [3] “Similar image search for histopathology: SMILY”, by N. Hedge et al.

---

> > ### Author Response · Authors · 2019-11-13
> > **Response to Review #3  - continued**
> >
> > Q3: The authors clarified that ProtoAttend is an inherently interpretable algorithm. However, the interpretability is proved by naïve Prototype learning only. A rigorous theoretical proof should be provided to demonstrate its interpretability.
> >
> > A3: We have added a new section to the Appendix, “Relation to Influence Functions”, that sheds light on the relation of ProtoAttend to Influence Functions that were proposed based on the theoretical insights on the machine learning loss functions. We hope that this Section can help contribute the additional theoretical rigor requested for our method.
> >
> > We also thought about whether it would be possible to provide even more rigorous theoretical proofs.  More specifically, we claim that ProtoAttend is inherently-interpretable based on sample-based interpretability - in other words, for each sample, it explains its output decisions by showing the most relevant prototypes. We encourage this goal with the listed principles in Section 3. If these principles are satisfied, in an ideal method we would obtain the desired interpretability - the coefficients of each prototype would correspond to how much they contribute to the output because of the linear combination of the values that ideally extracts all the relevant information from the input data. This follows by design.
> >
> > However, for the actual practical implementation, rigorous theoretical proofs are more difficult due to the implementation involving deep neural network architectures and loss functions for which theoretical analysis is challenging and limited except in idealized or simplified scenarios (as in many other deep learning research directions). Below we explore a theoretical analysis for one such simplified scenario:
> >
> > Let’s initially consider the scenario of convergence to the zero loss with \alpha=0 and \alpha=1. In this scenario, on training dataset, the value vector of the inputs would perfectly contain the relevant information for decision making, and the linear combination of the value vectors for the prototypes would perfectly contain the relevant information for decision making for the inputs. Moreover, in this linear combination, relevance is directly given by the coefficient. When the performance on test dataset is considered, the distribution mismatch would determine how much deterioration there would be in encoding the inputs and reasoning from the prototypes. In the absence of any generalization gap, we would obtain an error-free model with inherent sample-based interpretability. As test dataset deviates more from the training dataset, the classification performance would drop for decision making with \alpha=0 as well as decision making with \alpha=1.
> >
> > As we have non-zero loss, empirical errors are introduced both in how the encoder extracts the relevant information from the input, as well as how much the linear combination of prototypes value vectors would match to the inputs value vector, as a way of quantifying relevance. We empirically show in Table 1 that our design choices are still highly effective in addressing these, and indeed we can obtain similar performance to the baseline model (\alpha=0), as well as similar performance when decision making from the linear combination of prototype value vectors are considered (\alpha=1). These suggest that the generalization penalty mostly occurs where we encode the input data to value representations, not how we consider representations in inherently-interpretable prototypical learning setting.
> >
> > Overall, we can quantify various metrics (as in Table 1) as a way of estimating how much the desired principles are satisfied empirically, but a really rigorous theoretical proof would rely on generalization and convergence properties of deep neural networks that have not matured enough yet to be able to be applied to our setting.  We hope the reviewer can understand this.  In addition to the new theoretical analysis section, we will also add parts of the above discussion to the paper.
> >
> > Q4: I would be appreciated if the authors provide the pseudo-code to show the training procedure of ProtoAttend.
> >
> > A4: This is a great suggestion to improve readability. We have included the pseudo-code of training in Appendix A.
> >
> > Q5: Overall, I think this work is not ready for publishing unless the theoretical property is well understood.
> >
> > A5: We hope that we have made our contributions more clear, and provided convincing theoretical analyses and discussions. We hope that given our novel ideas, contributions and strong empirical results, this paper can constitute a basis for future papers on impactful applications and theoretical foundations. Please let us know if you have further questions.

---

### Official Review · AnonReviewer1 · 2019-10-23
**Official Blind Review #1**

**Rating:** 6

**Review:**

The aim of this work is to make deep learning classifiers more interpretable by "projecting" each input sample into a small collection of prototype examples (with some weighting over those) and then basing the decision on a combination of the latent representations of the chosen prototypes. In this way, the chosen category can be justified as the input being similar to the selected prototypes. Additionally, this approach makes it possible to obtain a confidence score at test time.

The choice fo the encoders for the prototypes and the examples is asymmetric (the first using keys and the second queries). This is not justified. Is it empirically better than using the same encoding for both before feeding them to the relational attention?

This work aims to satisfy many desiderata (listed in section 3). The decisions made to accomplish these are reasonable although somewhat arbitrary. In fact, several ways to encode the desiderata in the loss function are listed in Table 1.

Qualitatively, in the presented comparison with influence networks and representer point selection, ProtoAttend seems to choose more representative examples.

It is not easy to find a direct, quantitative way to compare this type of work with the existing literature, but from a qualitative perspective, the set goal (which is an important one) is achieved.



**Experience Assessment:**

I do not know much about this area.

**Review Assessment: Checking Correctness Of Derivations And Theory:**

I assessed the sensibility of the derivations and theory.

**Review Assessment: Checking Correctness Of Experiments:**

I assessed the sensibility of the experiments.

**Review Assessment: Thoroughness In Paper Reading:**

I read the paper at least twice and used my best judgement in assessing the paper.

---

> ### Author Response · Authors · 2019-11-14
> **Response to Review #1**
>
> Thanks for your valuable comments overall and specific suggestions below which helped us to improve the quality of our paper.
>
> Q1: The choice of encoders for the prototypes and the examples is asymmetric (the first using keys and the second queries). This is not justified. Is it empirically better than using the same encoding for both before feeding them to the relational attention?
>
> A1: Overall, we previously considered designs where encoders are not shared between the inputs, as well as using the same vectors for keys and queries as the output of the shared encoder. Empirically, we have observed that our current approach yields slightly superior performance and faster convergence.
>
> It is preferable to have the vast majority of parameters shared between the encoders for the inputs and the prototype candidates, as they both come from the same input data distribution and parameter sharing allow more efficient utilization of learning capacity and faster convergence. Keys, queries, and values correspond to different types of information. Queries summarize the information on how a particular input should be related to the candidate samples, keys summarize the information on how a particular candidate sample should be related to the input, and values summarize the content of the input or candidate data. Using the same values is important for the input samples and the candidate samples, because we would like to share the decision space to encourage principles, as given by the explanations followed by Eq. (2). For keys and queries, it makes sense to have them different, because the entire system is not symmetric, there are a lot of candidate samples and the model may prefer to learn the keys to arrange the representation space such that it is meaningful when their inner products with a single query are considered. Many deep learning applications where an attention mechanism is employed, particularly those with self-attention, map separate keys, queries and values from the same representation, as the inner product operation to determine alignment has more capacity when there is no symmetry requirement.
>
> We have added some of these discussions to the paper and validated that these design choices yield better empirical performance.
>
> Q2: This work aims to satisfy many desiderata (listed in section 3). The decisions made to accomplish these are reasonable although somewhat arbitrary. In fact, several ways to encode the desiderata in the loss function are listed in Table 1.
>
> A2: We first present the desiderata and then explain particular design choices to implement them efficiently. There may be other reasonable design choices for the loss functions and the model architecture that are not given in our paper and that can still implement desiderata well. Particularly with the loss functions, our goal is to show that a simple loss function that is superposition of two or three terms is already sufficient to encourage these principles.
>
> Q3: Qualitatively, in the presented comparison with influence networks and representer point selection, ProtoAttend seems to choose more representative examples.
>
> It is not easy to find a direct, quantitative way to compare this type of work with the existing literature, but from a qualitative perspective, the set goal (which is an important one) is achieved.
>
> A3: We appreciate that you find that we have achieved the important set of goals! We hope that we have fully addressed your concerns. Please let us know if you have further questions.

---

### Official Review · AnonReviewer2 · 2019-10-26
**Official Blind Review #2**

**Rating:** 3

**Review:**

This paper presents a sample-based self-explaining method for image classification. The basic idea is adopt the attention mechanism to learn the relation between the latent representation of the query sample and training samples, and identify the training samples with higher similarity as the prototype. The classification decision is based on the label consistency between the identified prototypes (with the relation score in attention mechanism as the weight of different prototypes in determining the  label agreement)

The proposed model is intrinsically interpretable since the prototypes with higher weights can play as the decision explanation. And the authors have conducted experiments to show that such self-explaining mechanism based on attention model can achieve comparable classification accuracy with original black-box models.

The presentation of the paper is clear and easy to follow. But I have several concerns regarding the choice of prototypes and the evaluation of the interpretation:

1) According to Eq. (2), it seems that all training samples are used as the prototypes (but with different weights). Why not just use the top few prototypes? Would this such setting introduce a lot of noise, since many training samples are from different classes?

2) Since one focus of the paper is to provide interoperation of the classification model, some more experiments are needed to evaluate how well the interoperation is. For example, some crowdsourcing experiments to check if the provided prototypes can help human users correctly guess the model prediction.

3) I think the authors should also compare with the black-box model when we use the attention mechanism as a post-hoc interpretation. One straightforward baseline is that use the black-box model for classification, and pick the top "prototypes" with the highest similarity in the latent representation. Such comparison can help to validate if incorporating attention mechanism in the model design can provide better quality prototypes.

**Experience Assessment:**

I have read many papers in this area.

**Review Assessment: Checking Correctness Of Derivations And Theory:**

I assessed the sensibility of the derivations and theory.

**Review Assessment: Checking Correctness Of Experiments:**

I assessed the sensibility of the experiments.

**Review Assessment: Thoroughness In Paper Reading:**

I read the paper at least twice and used my best judgement in assessing the paper.

---

> ### Author Response · Authors · 2019-11-15
> **Response to Review #2**
>
> Thanks for your valuable comments overall, and finding that we have achieved the intrinsically interpretable model design goal and presented it in a clear way.  See below for answers to questions as well as the requested experiments.
>
> Q1: According to Eq. (2), it seems that all training samples are used as the prototypes (but with different weights). Why not just use the top few prototypes? Would this such setting introduce a lot of noise, since many training samples are from different classes?
>
> A1: Thanks for pointing this out – we try to clarify this below:
>
> 1) The database size D is the size of the database of prototype candidates, and D may be as large as the entire training dataset at inference but that is not necessary. During training, at each iteration, we randomly sample a batch of prototype candidates, so in practice D is a small subset of the training dataset (see Table 4 for numerical values). The size of the prototype candidate batch should be sufficiently large such that the model can attend to reasonable prototypes with high coefficients (separately for each input).
>
> 2) The impact of each prototype in decision making is proportional to its coefficient because of the linear combination, and with appropriate sparsity mechanisms, we normally only end up with a few prototypes with large coefficients. Indeed, most of the coefficients would be zero with sparsemax activation and sparsity regularization.
>
> 3) For sparsity regularization, instead of the entropy term, we could alternatively hard-threshold coefficients to a few during training.  This would force the network to only use the top few prototypes, but we observed issues in converges with this direction, mostly because zeroing the gradient contributions from many samples would throw away valuable information.  This is why similarly to most other applications of deep learning we instead use soft approximations to selection operations.
>
> We have added parts of this discussion to the paper.
>
> Q2: Since one focus of the paper is to provide interoperation of the classification model, some more experiments are needed to evaluate how well the interoperation is. For example, some crowdsourcing experiments to check if the provided prototypes can help human users correctly guess the model prediction.
>
> A2: This is a great suggestion — we have added a user study to Appendix F. Even in this simple study that is not conducted on experts of the classification task and in which scores are averaged out due to human subjectiveness, ProtoAttend (4.33) beats the random (1.33) and random same-class (3.97) baselines by a significant margin.  We leave further and more comprehensive evaluation of human interoperation to future work, as it might be more meaningful to study it in the specific use case scenario (e.g. medical diagnosis) with a variety of domain-specific metrics.
>
> Q3: I think the authors should also compare with the black-box model when we use the attention mechanism as a post-hoc interpretation. One straightforward baseline is that use the black-box model for classification, and pick the top "prototypes" with the highest similarity in the latent representation. Such comparison can help to validate if incorporating attention mechanism in the model design can provide better quality prototypes.
>
> A3: Our experiments on Animals with Attributes, shown in Fig. 7, provide this comparison. We obtain the encoded representations of images from a pre-trained black-box model, and then apply our attention mechanism with a shallow encoder on them and observe that we observe better quality prototypes compared to alternative techniques, as a way of post-hoc explainability.
>
> We have also added a new section to the Appendix, “Relation to Influence Functions”, where we provide extra explanations on the comparison on the methods for similarity determination in the latent representation – we hope that is helpful!
>
> We hope that we have fully addressed your concerns. Please let us know if you have further comments.

---

### Decision · Program_Chairs · 2019-12-19

**Decision:**

Reject

**Comment:**

This paper proposes an interpretable machine learning method, ProtoAttend, that bases decisions on few relevant "prototypes." The proposed method uses an attention mechanism (possibly sparse, via sparsemax) that relates the encoded representations to samples in order to determine prototypes. The resulting model enables similarity-based interpretability, confidence estimation by quantifying the mismatch across prototype labels, and can be used for distribution mismatch detection.

While the proposed model is interesting, the reviewers raised several concerns regarding the choice of prototypes and the evaluation of human interoperation. The paper would benefit from more experiments besides the provided user studies to check if the provided prototypes can help human users correctly guess the model prediction. I encourage the authors to address these suggestions in a future resubmission.